



# Small-molecule inhibitors of the PDZ domain of Dishevelled proteins interrupt Wnt signalling

Nestor Kamdem [1,2], Yvette Roske [3], Dmytro Kovalskyy [4,6] Maxim O. Platonov [4,6], Oleksii Balinskyi [4,6], Annika Kreuchwig [1,2], Jörn Saupe [1,2], Liang Fang [2,3], Anne Diehl [1], Peter Schmieder [1], Gerd Krause [1], Jörg Rademann [1,5], Udo Heinemann [2,3], Walter Birchmeier [3] and Hartmut Oschkinat [1,2].

[1] Leibniz-Forschungsinstitut für Molekulare Pharmakologie, Robert-Rössle-Straße 10, 13125 Berlin, Germany
[2] Institut für Chemie und Biochemie, Freie Universität Berlin, Takustraße 3, 14195 Berlin, Germany
[3] Max-Delbrück-Center for Molecular Medicine, Robert-Rössle-Straße 10, 13125 Berlin, Germany
[4] Enamine Ltd., Chervonotkatska Street 78, Kyiv 02094, Ukraine
[5] Institut für Pharmazie, Freie Universität Berlin, Königin-Luise-Straße 2 + 4, 14195 Berlin, Germany
[6] Taras Shevchenko National University, 62 Volodymyrska, Kyiv 01033, Ukraine

*Correspondence to:* Hartmut Oschkinat (oschkinat@fmp-berlin.de)

**Abstract**

Dishevelled (Dvl) proteins are important regulators of the Wnt signalling pathway, interacting through their PDZ domains with the Wnt receptor Frizzled. Blocking the Dvl PDZ/Frizzled interaction represents a potential approach for cancer treatment, which stimulated the identification of small molecule inhibitors, among them the anti-inflammatory drug Sulindac and Ky-02327. Aiming to develop tighter binding compounds without side effects, we investigated structure-activity relationships of sulfonamides. X-ray crystallography showed high complementarity of anthranilic acid derivatives in the GLGF loop cavity and space for ligand growth towards the PDZ surface. Our best binding compound inhibits Wnt signalling in a dose-dependent manner as demonstrated by TOP-GFP assays (IC$_{50}$ ~50 μM), and Western blotting of β-catenin levels. Real-time PCR showed reduction in the expression of Wnt-specific genes. Our compound interacted with Dvl-1 PDZ (K$_d$=2.4 μM) stronger than Ky-02327 and may be developed into a lead compound interfering with the Wnt pathway.

KEYWORDS: Drug Design, NMR, PDZ, Frizzled, Wnt signalling



**INTRODUCTION**


Post synaptic density protein (PSD95), Drosophila disc large tumour suppressor (Dlg1), and Zonula
occludens-1 protein (ZO-1) domains termed PDZ appear in 440 copies spread over more than 260
proteins of the human proteome (Ponting 1997). They maintain relatively specific protein-protein
interactions and are involved, for example, in signalling pathways, membrane trafficking and in the
formation of cell-cell junctions (Zhang 2003, Fanning 1996, Kurakin 2007). Hence, they are potentially
attractive drug targets (Rimbault 2019, Christensen 2020). PDZ domains consist of about 90 amino acids
which fold into two α-helices and six β-strands exposing a distinct peptide-binding groove (Doyle 1996),
Lee 2017). The conserved carboxylate-binding loop (GLGF loop) is essential for the formation of a
hydrogen bonding network between the PDZ domain and PDZ-binding, C-terminal peptide motifs, in
most cases coordinating the C-terminal carboxylate group of the interaction partner. In the respective
complexes, the C-terminal residue of the ligand is referred to as $P_0$; subsequent residues towards the N-
terminus are termed $P_{-1}$, $P_{-2}$, and $P_{-3}$ etc. Previous studies have revealed that $P_0$ and $P_{-2}$ are most critical
for PDZ-ligand recognition (Songyang 1997, Schultz 1998).
PDZ domains are divided into at least three main classes on the basis of their amino acid preferences at
these two sites: class I PDZ domains recognize the motif S/T-X-Φ-COOH (Φ is a hydrophobic residue
and X any amino acid). Class II PDZ domains recognize the motif Φ-X-Φ-COOH, whereas class III
PDZ domains recognize the motif X-X-COOH. However, some PDZ domains do not fall into any of
these specific classes (Pawson 2007, Sheng 2001, Zhang 2003). The Dvl PDZ domains, for example,
recognize the internal sequence (KTXXXW) within the frizzled peptide 525(GKTLQSWRRFYH)536
($K_D \sim 10$ µM) (Wong 2003, Chandanamali 2009).
Dishevelled proteins are modular proteins comprising 500 to 600 amino acids and containing three
conserved domains: an N-terminal DIX (**Di**shevelled/A**x**in) domain, a central PDZ domain, and a C-
terminal DEP (**D**ishevelled, **E**gl-10 and **P**leckstrin) domain (Wong 2000, Wallingford 2005). They
transduce Wnt signals from the membrane receptor Frizzled to downstream components *via* the
interaction between Dvl PDZ and Frizzled (Wong 2003), thus it has been proposed as drug target (Klaus
2008, Holland 2013, Polakis 2012). Several studies identified internal peptides of the type (KTXXXW)

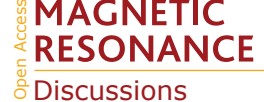

as well as C-terminal peptides of the type (ΩΦGWF) in which Ω is any aromatic amino acid (F, W or
Y) as Dvl PDZ targets (Lee 2009, Zhang 2009). Three Dvl homologues, Dvl-1, Dvl-2 and Dvl-3, have
been identified in humans and are highly conserved. The sequence identity is 88% between Dvl-3 PDZ
and Dvl-1 PDZ and 96% between Dvl-3 PDZ and Dvl-2 PDZ (Supporting Information Figure S1). Dvl
proteins are found to be upregulated in breast, colon, prostate, mesothelium, and lung cancers
(Weeraratna 2002, Uematsu 2003, Uematsu 2003, Bui 1997, Mizutani 2005). There are several
examples of small-molecule inhibitors of Dvl PDZ. NSC668036 (Shan 2005, Wang 2015) is a peptide-
mimic compound which interferes with Wnt signalling at the Dvl level. Based on a computational
pharmacophore model of NCS668036, additional compounds were later reported (Shan 2012). Known
as first non-peptide inhibitor, the 1H-indole-5-carboxylic acid derivative FJ9 (Fujii 2007) showed
therapeutic potential. Further examples including Sulindac (Lee 2009), 2-((3-(2-
Phenylacetyl)amino)benzoyl)amino)benzoic acid (3289-8625, also called CalBioChem(CBC)-322338)
(Grandy 2009, Hori 2018), N-benzoyl-2-amino-benzoic acid analogs (Hori 2018), phenoxyacetic acid
analogs (Choi 2016) , and Ethyl 5-hydroxy-1-(2-oxo-2-((2-(piperidin-1-yl)ethyl)amino)ethyl)-1H-
indole-2-carboxylate (KY-02327) (Kim 2016) have been reported, with the latter showing the highest
*in-vitro* affinity (8.3 µM) of all. Despite the existence of the abovementioned inhibitors of Dvl PDZ, the
development of tighter-binding, non-peptidic small-compound modulators of the respective functions,
binding with nanomolar affinity, is necessary and remains challenging. Here, nuclear magnetic
resonance (NMR) spectroscopy was used to detect primary hits and for follow-up secondary screening.
The ability of NMR to detect weak intermolecular interactions ($\mu M < K_D < mM$) make it an ideal
screening tool for identifying and characterizing weakly binding fragments, to be optimized
subsequently by chemical modification in order to improve binding (Zartler 2006, Shuker 1996, Zartler
2003). Besides NMR, the determination of X-ray crystal structures of selected complexes was
fundamental for further design of new compound structures with improved binding. In the first round of
screening, a library constructed after computational docking of candidates into the peptide binding site
of the Dvl PDZ domains were investigated, followed by secondary screening utilizing a library of 120
compounds containing rhodanine or pyrrolidine-2,5-dione moieties.
**RESULTS AND DISCUSSION**

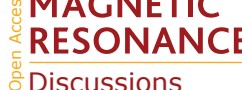

**PDZ targeted library design**

The PDZ targeted library was designed to cover all PDZ domains with available structure. For this, all X-ray and NMR derived PDZ structures were retrieved from the PDB, clustered, and 6 selected centroids were subjected to the virtual screening routine. The clustering of the PDZ domains was performed according to the shapes of their binding sites, rather than backbone conformation. This approach accounts for the importance of surface complementarity of protein-small molecule interactions and the critical contribution of van der Waals interactions to the binding free energy. On another hand, PDZ domains have evolved to recognize a carboxyl group that is mostly derived from the C-terminus of natively binding proteins. Finally, the fact that PDZ can recognize internal motifs (Hillier 1999), including KTXXXW of Frizzled-7 recognised by Dvl PDZ (Wong 2003, Chandanamali 2009), raises the question of what are key binding contributions with PDZ domains: negative charge, hydrogen bonding or shape complementarity (Harris 2003). For this reason, tangible compounds were preselected to have extensive hydrophobic contacts as well as chemical groups that mimic the carboxylic group.

Virtual screening was performed with QXP, and the generated complexes were sequentially filtered with a self-designed MultiFilter algorithm. From the resulting 1119 compounds a randomly selected set of 250 compounds was subjected to NMR validation.

**NMR Screening and development of compounds**

The results of virtual screening were checked experimentally by comparing 2D $^1$H-$^{15}$N HSQC (Heteronuclear Single Quantum Correlation) spectra of Dvl-3 PDZ in the absence and presence of the compound to elucidate ligand-induced changes of chemical shifts. Chemical shift perturbation differences ($\Delta$CSPs) were evaluated for compounds that cause shifts of at least three N-H cross-peaks. The responses were classified into: (i) inactive compounds ($\Delta$CSP < 0.02); (ii) very weak interactions (0.02 ≤ $\Delta$CSP ≤ 0.05); (iii) weak interactions (0.05 < $\Delta$CSP ≤ 0.1); (iv) intermediate interactions (0.1 < $\Delta$CSP ≤ 0.2); (v): strong interactions (0.2 < $\Delta$CSP ≤ 0.5) and (vi) very strong interactions ($\Delta$CSP > 0.5). In most cases, the signals of residues S263, V287 and R320 were perturbed (Supporting Information Figure S2). With the $\Delta$CSP of 0.12 ppm, the isoleucine-derived compound **1** ((2,3-dihydrobenzo[b][1,4]dioxin-6-yl)sulfonyl)-L-isoleucine containing a sulfonamide moiety was detected





initially as one of the best "hits" according to chemical shift changes. The sulfonamide is a well-known
moiety in drug discovery (Mathvink 1999, Wu 1999, Sleight 1998 O'Brien 2000, Tellew 2003).


**1**

Upon NMR titration experiments for compound **1** (Supporting Information Figure S2) with Dvl-3 PDZ,
the largest chemical shift perturbations were observed for S263 in strand βB and R320 in helix αB of
Dvl-3 PDZ, confirming the conserved binding site.

        **2**                    **3**                    **4**                    **5**

**Scheme 1**: Compounds **2**, **3**, **4**, **5**
By comparing the binding of several sulfonamide compounds in a secondary screening event and mak-
ing use of our in-house library, four new compounds (**2**, **3**, **4**, **5**) that induced chemical shift perturbations
larger than 0.2 ppm were found (for binding constants see Table 1) and considered further as reasonably
strong binders. The similarity of the structures led us to define Scheme 2 as a scaffold for further refine-
ments. Sulfonamides were considered more drug-like, and hence followed up at higher priority than
other hits. We realised that our four new compounds had different moieties at $R_2$ in combination with a
small $R_1$ (fluorine). A decrease of binding was observed with decreasing size of $R_2$.






**Scheme 2**: Basic fragment for further synthesis
In order to assess the importance of the aryl group at $R_2$ for complex formation, it was replaced by a
methyl group as substituent to yield compound **6**, which showed a drastic decrease of binding (Table 1).
Compounds **3**, **4**, and **5** did not distinguish between the Dvl-3 PDZ and Dvl-1 PDZ. In order to obtain
detailed insight into the binding mode of these compounds, crystal structures of Dvl-3 PDZ in complex
with compounds **3**, **5** and **6** were determined (Figure 1). For compound **3** the crystal structure revealed
two complexes within the crystallographic asymmetric unit (AU) at 1.43 Å resolution. Both show the
anthranilic acid with the attached fluorine pointing into the hydrophobic binding pocket (Figure 1A and
Supporting Information Figure S3A), while the carboxyl group forms a hydrogen-bond network with
amide residues of the carboxylate binding loop, in particular strand βB (Figure 1A) and specifically with
residues I262, G261 and L260. The two sulfonamide oxygen atoms form hydrogen bonds with R320
and H324 (weak) of helix αB for only one complex in the AU. The aromatic aryl group
(tetrahydronaphtalene) attached to the sulfonamide is involved in hydrophobic interactions with F259
(Supporting Information Figure S3B). The 1.6-Å complex structure with compound **5** (4 molecules per
AU) exhibits a comparable binding mode as found for compound **3** with a hydrogen-bond network
involving the carboxyl group and the amides of I262, G261, L260, and of the sulfonamide to H324
(Figure 1B). No hydrogen bond was observed to R320 in all four molecules of the AU, but small
variations of the aryl moiety relative to F259 (Supporting Information Figure S3C). The crystals of the
complex with **6** show two molecules in the AU (Figure 1C). The sulfonamide is bound by H324 in both
complexes (Supporting Information Figure S3D). However, compound **6** bound only in the mM range
as compared to **3** and **5**, which obviously results from the missing aromatic rings.



| ID | R₁ | R₂ | (K_D, μM) Dvl-3PDZ | (K_D, μM) Dvl-1 PDZ |
|---|---|---|---|---|
| 2 | F | *tetralin* | nd | $237.6 \pm 38.5^{NMR}$ |
| 3 | F | *indane* | $80.6 \pm 6.1^{NMR}$ | $112.7 \pm 25.9^{NMR}$ |
| 4 | F | *indane* | $83.9 \pm 7.8^{NMR}$ | $114.4 \pm 9.8^{NMR}$ |
| 5 | F | *dimethylbenzyl* | $140.6 \pm 14.1^{NMR}$ | $160.1 \pm 14.6^{NMR}$ |
| 6 | F | CH₃ | $> 1000^{ITC}$ | - |
| 7 | Br | *tetralin* | $20.6 \pm 2.4^{NMR}$ | $18.2 \pm 2.4^{NMR}$ |
| 8 | CF₃ | *tetralin* | $17.4 \pm 0.5^{ITC}$ | $24.5 \pm 1.5^{ITC}$ |
| 9 | Cl | *tetralin* | $41.1 \pm 3.1^{NMR}$ | $45.6 \pm 4.5^{NMR}$ |
| 10 | CH₃ | *tetralin* | $62.5 \pm 4.7^{NMR}$ | $60.5 \pm 5.3^{NMR}$ |
| 11 | Br | *naphthalene* | $13.8^{ITC}$ | $119.9^{ITC}$ |
| 12 | Br | *benzyl* | $58.5^{ITC}$ | nd |
| 13 | Br | *phenoxybenzyl* | $7.2^{ITC}$ | $213.2^{ITC}$ |
| 14 | Br | *trimethylbenzyl* | $58.1 \pm 2.1^{ITC}$ | nd |
| 15 | CF₃ | *acetylbenzyl* | $52.9 \pm 1.7^{ITC}$ | nd |
| 16 | CF₃ | *benzodioxane* | $59.1 \pm 1.5^{ITC}$ | nd |
| 17 | CF₃ | *trimethylbenzyl* | $49.5^{ITC}$ | nd |
| 18 | CH₃ | *indazole carboxamide (chloro)* | $9.4 \pm 0.6^{ITC}$ | $2.4 \pm 0.2^{ITC}$ |
| 19 | CH₃ | *indole carboxamide (chloro)* | $21.8 \pm 1.7^{ITC}$ | $8.0 \pm 0.5^{ITC}$ |
| 20 | CH₃ | *bromopyrrole carboxamide (chloro)* | $9.8 \pm 0.3^{ITC}$ | $4.7 \pm 0.3^{ITC}$ |
| 21 | CH₃ | *chloropyrrole carboxamide (chloro)* | $12.5 \pm 0.5^{ITC}$ | $4.7 \pm 0.2^{ITC}$ |

**Table 1.** Binding constants Kd (μM) of Dvl-3 PDZ and Dvl-1 PDZ for compounds 3 – 21 derived by ITC or NMR if not
specified.



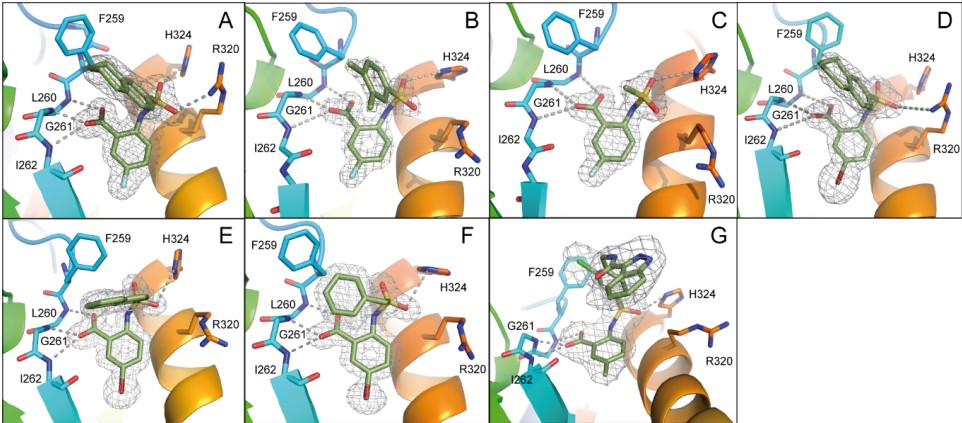



**Figure 1**: Magnified view into crystal structures of various compounds bound to the Dvl-3 PDZ domain. The 2Fo-Fc electron density around the compounds is shown at 1σ contour level, and the dotted lines indicate formed hydrogen bonds. In the bound compounds covalent bonds to carbon atoms are shown as green sticks. Important residues involved in compound binding are labelled and displayed in atom colours (carbons blue or dark yellow). **A-C** show compound **3**, **5** and **6** respectively. All compounds in **A-C** contain fluorine (light blue) in para position to the amine. **D-F** represents the bound compounds **7**, **11** and **12**, respectively. All have bromine (dark red) in para position to the amine. The accession codes of the structures are 6ZBQ, 6ZBZ, 6ZC3, 6ZC4, 6ZC6, and 6ZC8.

To further explore the importance of the fluorine site inside the hydrophobic pocket, substitutions by
bromine, chlorine, methyl and trifluoromethyl were chosen. In fact, the methyl group has a similar vdW
radius as the CF$_3$ group. Iodine was not considered a good candidate since it increases molecular weight
substantially and the compounds may be chemically less stable, in particular in biological assays. Taking
into account that compound **6** did not bind because of the missing aromatic ring at the R$_2$ position, our
strategy was to increase the aromatic ring at R$_2$ while keeping R$_1$ as small as possible (preferably CH$_3$)
to enable further compound modifications that fulfil key properties as defined by Lipinski (Lipinski
2000, Lipinski 1997). Our preference to continue exploration only at the R$_1$ position of the aromatic ring
in Scheme 1 was inspired by the absence of hits with other substitutions in the secondary screening event
and the initial X-ray structures that showed a hydrophobic pocket available for substituents in this
position while other sites at the aromatic ring would include steric hindrance. Therefore, compounds **7-**
**17** were obtained and were classified in three different groups to derive structure activity relationships
(SAR). The compounds **7-10** in group 1 contain different R$_1$ (Br, CF$_3$, Cl, CH$_3$) but the same moiety
(tetrahydronaphthalene) at R$_2$. As expected, binding could be further improved by displacement of the
fluorine with elements exhibiting larger van der Waals (vdW) radii. Indeed, the K$_D$ decreased stepwise
and the best fit was observed for compound **8** containing a trifluoromethyl group (K$_D$ = 17.4 μM for

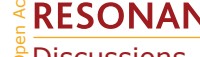

Dvl-3 PDZ and 24.5 µM for Dvl-1 PDZ). The different substituents at the $R_2$ position contribute to an
increased binding affinity in the following order: F < Cl < Br < $CF_3$ (compound **3** < **9** < **7**< **8**,
respectively). Compound **10** with a methyl group at the $R_2$ position showed only marginally improved
binding, although the methyl group has a similar vdW radius as the $CF_3$ group of compound **8**. The
difference in binding results most likely from their different hydrophobicity.
The 1.85-Å crystal structure of the Dvl-3 PDZ domain with compound **7** ($K_D = 20.6$ µM for Dvl-3 PDZ
and 18.2 µM for Dvl-1 PDZ) showed an identical hydrogen-bond network involving the amide groups
of residues I262, G261 and L260 of the carboxyl binding loop as seen for all other complex structures
reported here (Figure 1D). Only one hydrogen bond between the sulfonamide and R320 was found in
addition for one of the two Dvl-3 PDZ molecules per AU. H324 of Dvl-3 PDZ was not addressed by the
sulfonamide as seen previously. The bromine at position $R_1$ points into the hydrophobic pocket, similar
as the fluorine in the complex structure with compound **3**. The two complexes in the AU show significant
variations in the positions of the tetrahydronaphtalene rings as well as for the side chain of F259 and
R320 (Supporting Information Figure S3E).
Following the analysis of the complex involving compound **7**, the binding characteristics of the group-
2 compounds (**11-14**) were investigated. They contain bromine as $R_1$ and different substituents at the $R_2$
position to assess the importance of π-π stacking interactions involving F259. $K_D$ values of 7.2 µM for
compound **13** and 13.8 µM for compound **11** were found with respect to the interaction with Dvl-3 PDZ.
Crystal structures of Dvl-3 PDZ in complex with compound **11** (1.58 Å resolution, 1 molecule per AU)
and **12** (1.48 Å, 2 molecules per AU) revealed very similar binding as observed in the crystal structures
with compounds **3** and **7**. The aromatic rings at $R_2$ show hydrophobic interactions to F259, but not a
classical π-π stacking as expected. Nevertheless, the tighter binding of compound **11** could be explained
by the larger aromatic substituent at the $R_2$ position compared to compound **12**. Both complex structures
show also non-specifically bound ligands in crystal contacts (Supporting Information Figure S3H,
Supporting Information Tables S2 and S3). The additional ligand molecules in both complex structures
can be explained as a crystallographic artefact, which is verified with ITC experiments that indicate 1:1
stoichiometries in both cases (Figure S5). With respect to the selectivity of the tested compounds we
observed a 6 to 30-fold stronger binding of compounds **7**, **9**, **11** and **13** to Dvl-3 PDZ as compared to



Dvl-1 PDZ. These differences are related to the different sequences at the end of αB. Most importantly,
H324 is replaced by a serine residue in the Dvl-1 PDZ domain.
The group-3 compounds (15-17) contain a trifluoromethyl at position $R_1$ and were tested to investigate
a cooperative role of this moiety with various substituents at position $R_2$. All compounds bind weaker
to Dvl-1 and Dvl-3 than compound 8 which contains tetrahydronaphthalene at the $R_1$ position, revealing
its important role in the interaction.

**Further modifications towards higher affinity and reduced toxicity**
Possible cytotoxic effects of compounds 3, 7, 8, 9, and 10 were evaluated in cell viability assays using
HEK293 cells (Supporting Information Figure S4). These compounds were selected due to different
substituents at the $R_1$, including halogens. Cell viability was measured 24h after treatment with the
individual compounds, and half maximal inhibitory concentrations ($EC_{50}$) were calculated for each
compound. The compounds exhibited $EC_{50}$ values in the range of 61-131 μM (Supporting Information
Figure S4A). Compounds 3 and 10 that contained fluorine or methyl group substituents at $R_2$,
respectively, were the least toxic, while compound 7, containing bromine, was the most toxic. The
results from crystallography, modelling studies and of the cell proliferation assays led us to further
investigate compounds 18-21 that contain a methyl group at the $R_1$ position and different substituents
as $R_2$. In this way, we aimed to develop both potent and less toxic, cell permeable inhibitors. All
compounds showed strong interactions as indicated by chemical shift perturbation values between 0.30
to 0.34 ppm (Supporting Information Table S1). The binding constants were evaluated by ITC whereby
compound 18 ($K_D$ = 9.4 μM for Dvl-3 PDZ and 2.4 μM for Dvl-1 PDZ) appeared to be most potent.
Compound 18 contains a pyrazole ring which is considered as an important biologically active
heterocyclic moiety (Lv 2010). Compounds 20 ($K_D$ = 9.8 μM for Dvl-3 PDZ and 4.7 μM for Dvl-1 PDZ)
and 21 ($K_D$ = 12.5 μM for Dvl-3 PDZ and 4.7 μM for Dvl-1 PDZ) contain pyrrole rings. Their binding
constants almost have the same value despite the different substituents (bromine or chlorine) at the
pyrrole rings. The binding of compounds 18-21 to both Dvl PDZ domains is mainly enthalpy-driven as
indicated in Table 2, with a slightly stronger effect for Dvl-1 PDZ than for Dvl-3 PDZ. To our surprise,
the crystal structure of Dvl-3 PDZ in complex with compound 18 shows the pyrazole substituent in the



R$_2$ position orientated away from the binding pocket. Instead, a π-π stacking interaction with F259 was
observed (Supporting Information Figure S3I). Cytotoxicity of **18-21** was determined *via* MTT assays
(Mosmann 1983) that displayed viability up to concentrations above 150 μM (Supporting Information
Figure S4B).

| Compound | Dvl-3 PDZ | | | | Dvl-1 PDZ | | | |
|---|---|---|---|---|---|---|---|---|
| | Kd (μM) | ΔH (kcal/ mol) | TΔS (kcal/ mol) | ΔG (kcal/ mol) | Kd (μM) | ΔH (kcal/ mol) | TΔS (kcal/ mol) | ΔG (kcal/ mol) |
| **18** | 9.4 ± 0.6 | -8.0 | -1.2 | -6.8 | 2.4 ± 0.2 | -12.2 | -4.7 | -7.5 |
| **19** | 21.0 ± 1.7 | -5.9 | 0.4 | -5.5 | 8.0 ± 0.5 | -7.3 | -0.3 | -7.0 |
| **20** | 9.8 ± 0.3 | -10.4 | -3.6 | -6.8 | 4.7 ± 0.3 | -9.4 | -2.2 | -7.2 |
| **21** | 12.5 ± 0.5 | -5.9 | 0.7 | -6.8 | 4.7 ± 0.2 | -8.5 | -1.5 | -7.0 |
| NPL-1011 | 79.7 ± 53.3 | | | | | | | |
| Sulindac | 8.3 ± 2.5 | | | | | | | |
| CBC-322338/ 3289-8625 | > 400 μM | | | | | | | |
| NSC668036 | > 400 μM | | | | | | | |
| Ky-02327 | | | | | 8.3 ± 0.8[16g] | | | |

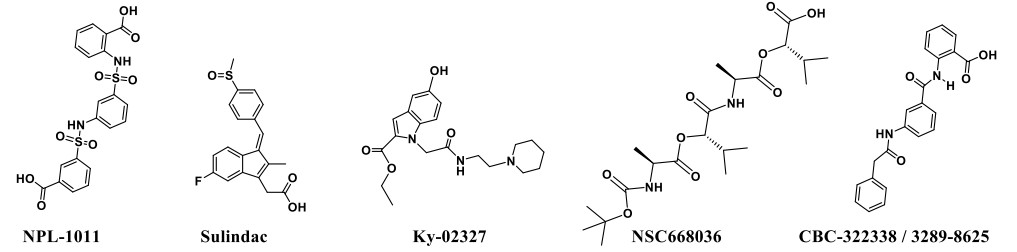


**Table 2:** Isothermal titration calorimetric data for the reaction between Dvl-3 PDZ, Dvl-1 PDZ and our compounds **18**, **19**, **20**
and **21** respectively. Compounds NPL-1011 (Hori 2018), and Sulindac (Lee 2009), CBC-322338/3289-8625 (Grandy 2009,
Hori 2018), and NSC668036 (Shan 2005), for more thermodynamic parameters see Supporting Information Figure S7. For Ky-
02327 the value from literature is included.

**Comparison to reported Dvl PDZ-binding molecules**
Our compounds bind to Dvl-3 with a Kd better than 10 μM, and slightly tighter to Dvl-1, see Table 2,
with **18** showing a Kd of 2.4 μM and chemical shift changes indicating binding to the canonical binding
site (Supporting Information Figure S8). For comparison, four compounds shown in Supporting
Information Figure S6 were assayed by ITC (Supporting Information Figure S7) regarding their affinity
to Dvl-3 PDZ. Ky-02327 was already determined to bind with a Kd of 8.3 ± 0.8 μM (Kim 2016) to Dvl-
1 PDZ. Our first interest was oriented towards sulfonamides. Hori et al (Hori 2018) have recently
reported 3-({3-[(2-carboxyphenyl)sulfamoyl]phenyl}sulfamoyl)benzoic acid (NPL-1011) binding to
Dvl-1 PDZ via the detection of chemical shift changes, and further sulfonamide compounds that showed
smaller effects, indicating weaker binding. We examined the binding constant of NPL-1011 which



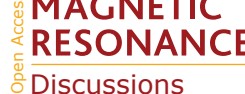

possesses two sulfonamide moieties by ITC and found a value of 79.7 ± 53.3 µM, see Table 2. For
further comparisons, we assayed also CBC-322338/3289-8625, Sulindac and NSC668036 by ITC.
Surprisingly, CBC-322338/3289-8625 showed very low affinity, with a Kd above 400 µM (assumed to
be the threshold for our ITC assay), which was larger than the originally reported value (10.6 +/- 1.7)
(Grandy 2009) and closer to the value found by Hori et al (Hori 2018) (954 +/- 403 µM). Concerning
non-sulfonamide compounds, a Kd of 8.3 ± 2.5 µM was detected for Sulindac, while NSC668036 (Shan
2005) did not show high-affinity binding. These results are largely in agreement with literature. In all
cases, compounds were tested for purity after Kd measurements (see Supporting Information Figures
S9A-D).

**Selectivity testing using a set of selected PDZ domains**
Compounds **18**, **20** and **21** were tested towards other PDZ domains for selectivity. The set included
PSD95-PDZ 2 and 3, Shank-3, α-syntrophin, and AF-6 PDZ. According to the determined chemical
shift perturbations (Supporting Information Table S4), our compounds show no or very weak
interactions with the selected PDZ domains (0.05 < ΔCSP ≤ 0.1) ppm. These findings led to the
conclusion that our compounds show considerable selectivity towards Dvl PDZ domains. This
selectivity might be due to a unique feature of Dvl PDZ where R320 (Dvl-3 PDZ) or R322 (Dvl-1 PDZ)
are crucial for interactions, explaining selectivity with respect to other PDZ domains. In addition, the
large hydrophobic cavity for the side chain of the C-terminal residue of the interacting peptide is
occupied by a large moiety in case of compounds **18**, **20** and **21** which might not be accommodated in
most other PDZ domains.

**Dvl inhibitors antagonize canonical Wnt signalling and Wnt-related properties of cancer cells**
Taking advantage of a lentivirus that encodes GFP in a β-catenin/TCF-dependent fashion (TOP-GFP,
SABiosciences), a stable HEK293 reporter cell line was established to evaluate the inhibitory effect of
compounds **18**, **20** and **21** on canonical Wnt signalling activity. TOP-GFP expression in this cell line
was induced by the ligand Wnt3a, which directly activates the Frizzled-Dishevelled complex and
protects β-catenin from degradation by the destruction complex (Figure 2A). Remarkably, all three
compounds inhibited Wnt signalling induced by Wnt3a in a dose-dependent manner (Figure 2B),



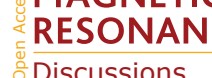

yielding $IC_{50}$ values between 50-80 µM. To further evaluate the specificity of our Dvl inhibitors, the
conventional TOPflash (Molenaar 1996) and other luciferase reporter assays were performed. In Hela
cells, **20** inhibited TOP-luciferase signals stimulated by Wnt3a but not by CHIR99021(Sineva 2010), a
compound that activates Wnt signalling downstream of Dvl (Figure 2A, C). Compound **20** had no
significant inhibitory effects in reporter assays that measure the activity of other signalling systems, e.g.,
NF-κB-luciferase stimulated by recombinant TNFα, Notch-luciferase stimulated by the overexpression
of the Notch intracellular domain, or the Oct-luciferase assay that is stimulated by overexpression of
Oct4 (SABiosciences, Figure 2D). These results strongly indicate that **20** is specific for canonical Wnt
signalling at the upstream level.
Increased β-catenin protein level is a hallmark of active Wnt signalling (Kishida 1999). Once β-catenin
is accumulated in the cytoplasm, it can translocate into the nucleus and activate the transcription of Wnt
target genes by interacting with transcription factors of the TCF/LEF family (Figure 2A) (Behrens
1996). In Hela cells, all three Dvl inhibitors blocked the increase of production of β-catenin by Wnt3a
in dose-dependent manners, as seen by Western blotting (Figure 2E). Increased mRNA



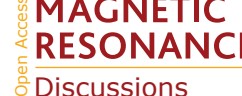

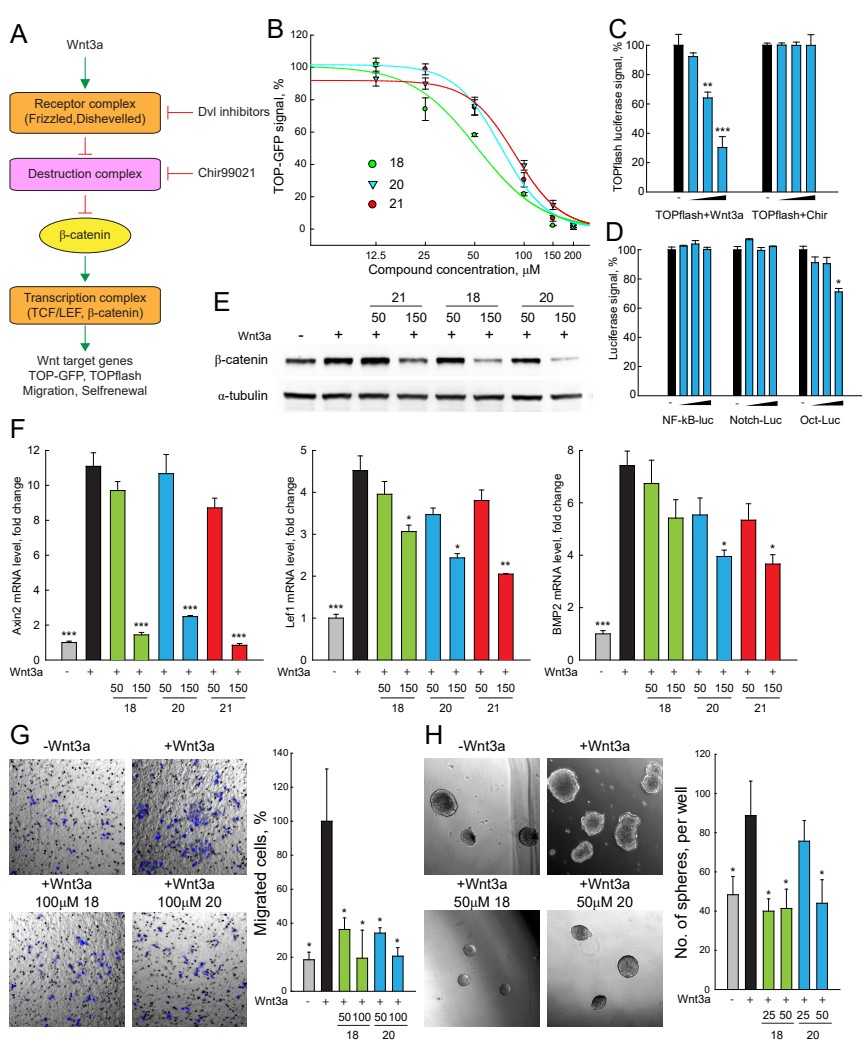

Figure. 2

**Figure 2.** DVL inhibitors antagonize Wnt signalling and Wnt related properties of cancer cells induced by Wnt3a. A. Scheme
of Wnt signalling pathway. Important components of the Wnt signalling pathway are schematically presented. Wnt3a treatment
increases the transcription of Wnt targets, enhances signals of TOP-GFP and TOPflash assays, and promotes Wnt related
biological properties of cancer cells. **B.** TOP-GFP reporter assays were performed with HEK293 reporter cell line. Compound
**18**, **20** and **21** inhibited Wnt3a induced Wnt activation in dose-dependent manner with $IC_{50}$ of 50~75 μM. **C&D.** TOPflash
assays stimulated with Chir99021 and reporter assays of other pathway were used to evaluate the specificity of compound **20**.
Compound **20** specifically inhibited Wnt3a induced Wnt activation, and had no or mild effect on Chir99021 induced Wnt



activation and other signalling pathways including NF-κB, Notch and Oct4. **E.** β-Catenin protein levels were detected with
Western blotting in Hela cells. Compound **18**, **20** and **21** (150 μM) inhibited accumulation of β-catenin in Hela cells treated
with Wnt3a. **F.** The mRNA levels of Wnt target genes (Axin2, Lef1 and Bmp2) in Hela cells were measured with quantitative
real-time PCR. Compounds **18**, **20** and **21** (150 μM) reduced the transcription of Wnt target genes that are enhanced by Wnt3a
treatment in Hela cells. **G.** Cell migration of SW480 cells after Wnt3a treatment was assessed by transwell assays. Compounds
**18** and **20** (50~100 μM) reduced the migration of SW480 cells enhanced by Wnt3a. **H.** SW480 cells were cultured in serum-
free non-adherent condition to evaluate the self-renewal property enhanced by Wnt3a treatment. Compound **18** and **20** (25~50
μM) reduced sphere formation of SW480 cells that was enhanced by Wnt3a treatment.

levels of the Wnt target genes Axin2, Lef1 and Bmp2 (Riese 1997, Jho 2002, Lewis 2010) were induced
by Wnt3a treatment, as measured by qRT-PCR, and these increases were reduced by compounds **18**, **20**
and **21** (Figure 2F). These results demonstrate that compounds **18**, **20** and **21** inhibit Wnt signalling as
indicated by reduced accumulation of β-catenin and low expression of typical Wnt target genes.
Canonical Wnt signalling contributes to cancer progression by inducing high motility and invasion of
cancer cells while retaining the self-renewal property of cancer initiating cells (Fritzmann 2009, Sack
2011, Vermeulen 2010, Malanchi 2008). In particular, cancer initiating cells are propagated and
enriched in non-adherent sphere culture, demonstrating the self-renewal capacity of the stem cells
(Kanwar 2010, Fan 2011). To investigate the potential value of the Dvl inhibitors for interfering with
these Wnt-related properties of cancer cells, the subline SW480WL was derived from the SW480 colon
cancer cell line, which exhibits a low level of endogenous Wnt activity (Fang 2012). The cell migration
and self-renewal properties of SW480WL cells were enhanced by Wnt3a treatment, as revealed by
transwell and sphere formation assays (Figure 2G, H). Compounds **18** and **20** prevented increased cell
migration and sphere formation. These results indicate that our Dvl inhibitors may be developed into
lead compounds that interfere with Wnt signalling.

**CONCLUSIONS**
In the present work, small molecules that bind to Dvl PDZ in the one-digit micromolar range with
considerable selectivity have been developed by an extensive structure-based design approach. With
regards to the affinity determined by ITC, compound **18** binds to Dvl-1 and Dvl-3 in vitro with Kd
values of 2.4 and 9.4, respectively, comparing very well with known ligands. X-ray structures of Dvl-3
PDZ complexes with selected compounds provided insight into crucial interactions and served as the
basis for the design of tight binding compounds with reduced toxicity. The structural investigations
revealed that these compounds form hydrogen bonds with the amide groups of residues L260, G261 and

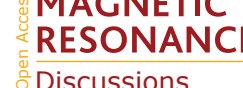

I262 in the PDZ-domain loop and the side chains of residues H324 and R320. Finally, the chosen
methodology, virtual screening followed by a two-stage NMR based screening, X-ray crystallography,
and chemical synthesis is an excellent path towards bioactive interaction partners. Our best compounds
effectively inhibited the canonical Wnt signalling pathway in a selective manner and could be developed
for further studies.

**Experimental Section**
**Clustering binding sites and selection of representative PDZ domains**
Three-dimensional structures of PDZ domains were retrieved from the PDB (Berman 2000). At the time
of the study from a total of 266 PDB files, 126 were NMR solution structures and 140 derived from X-
ray diffraction studies. The structures belong to 163 PDZ domains of 117 different proteins from 11
organisms. Files which contain more than one 3D conformation for a domain (up to 50 for NMR-derived
data) were split into separate structures and considered independently. The total number of unique 3D
structures was 2,708.
Amino acid sequences of PDZ domains were aligned using Clustal Omega software (Sievers 2011).
Based on the alignment, for each structure, residues which form the binding site (strand βB and helix
αB) were determined (Supporting Information Figure S8). The centre of the binding site was defined as
a geometric centre of Cα atoms of 7 residues (6 residues from the βB strand and the second residue from
the αB helix). Such bias toward the βB strand was made to cover sites occupied by residues in -1 and -
3 positions.
The triangulated solvent accessible surface for each PDZ structure was built using MSMS software
(Sanner 1996) with a spherical probe radius of 1.4 Å and vertex density 10 Å$^{-1}$. The largest connected
set of surface vertices within 9 Å from the centre of the binding site was used to construct shape-based
numerical descriptors. The descriptors are 508-dimensional vectors of non-negative integer numbers
and were built using a shape distributions approach (Osada 2002). In total 10 (Pawson 2007) vertex
triplets were selected randomly, each forming a triangle. Triangles which had a side longer than 16 Å
were discarded. Triangle sides were distributed into 16 length bins, each 1 Å wide, covering lengths
from 0 to 16 Å. A combination of three sorted side lengths, each belonging to one of 16 distance bins,



defines one of 508 categories of the triangles. The number of triangles of each category was calculated,
resulting in a 508-dimensional vector which is used as a numerical descriptor of the binding pocket
shape. For further operations with descriptors, Euclidian metric was introduced. Shape descriptors were
distributed into 6 clusters using k-means algorithm (Jain 1988). For each cluster, a centroid structure
was defined as the one, whose descriptor is the closest to mean descriptor for the cluster. The centroid
structures (2O2T#B.pdb, 1VA8#3.pdb, 2DLU#01.pdb, 1UHP#8.pdb, 2OS6#8.pdb, 3LNX#A.pdb) were
used for docking.

**PDZ targeted library design**
Screening collection by Enamine Ltd. (Chuprina 2010) containing a total of 1,195,395 drug-like
compounds was used as the primary source of small molecules. Natural ligand of PDZ is the C-terminus
of a peptide with carboxylic group making extensive hydrogen bond network with the "ΦGΦ" motif.
Since the carboxyl provides either of negative charge and hydrogen bond acceptor, we want our ligands
to retain at least one of these features. Therefore, we pre-filtered the stock library to bear chemical
groups which have negatively charged and/or hydrogen bond acceptor functionality. In total 65,288
compounds were selected for the virtual screening study. The selected 6 centroids of PDZ domains and
the prepared compound set were subjected to high-throughput docking using the QXP/Flo software
(McMartin 1997). Complexes were generated with 100 steps of sdock + routine, and 10 conformations
per complex were saved.
Processing of docking poses started with filtering by contact term *Cntc* from the QXP/Flo scoring
function. Entries with Cntc < 45 were discarded, which removed complexes with weak geometries of
bound ligands. The remained filtering was performed with the in-house MultiFilter program that allows
flexible geometry-based filtering. We applied two algorithms, *nearest-atom* filter and *hydrogen-bond*
filter. The former filters complexes by distance from a given protein atom to the nearest heavy ligand
atom, while in the latter, filtering is based upon the number of hydrogen bonds calculated for a given
complex geometry. With the *nearest-atom* routine we selected compounds that filled the $P_0$ pocket and
sterically mimicked binding of a peptide carboxylic group. Peptide group hydrogens of the "ΦGΦ" motif
and atoms forming the hydrophobic pocket were used for that. With the *hydrogen-bond* filter we selected

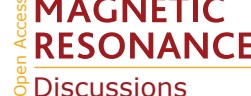

compounds that formed extensive hydrogen bonding with the PDZ domain. Both these properties might
have larger impact on binding rather than negative charge (Harris 2003). Details on atoms used for
filtering and thresholds for *hydrogen-bond* filtering, as well as the resulting number of compounds, are
provided in Supporting Information Table S5. Compounds from complexes which passed through these
filters were incorporated into a targeted library for the PDZ-domain family. The final library contained
1119 compounds in total.

**Screening of compounds**
Two-dimensional $^1$H-$^{15}$N HSQC spectra were used to screen a library of 212 compounds designed by
the company Enamine for PDZ domains. 50 μM of $^{15}$N-labeled protein samples were prepared in a 20
mM sodium phosphate buffer, containing 50 mM sodium chloride, 0.02% (w/v) NaN$_3$, at pH 7.4. Stock
solutions of small molecules were prepared in DMSO-*d6* at a concentration of 160 mM. A $^1$H-$^{15}$N HSQC
spectrum of Dvl PDZ was acquired at 300 K with 5% DMSO-*d6* in the absence of ligand as reference
spectrum. Mixtures of 16 compounds were added to $^{15}$N-labeled Dvl PDZ at 8-fold molar excess each.
The final concentration of DMSO-*d6* in the protein-ligand solutions was 5%. Spectra were acquired
with 8 scans and 256 points in the indirect dimension. Compound binding was deduced if the resonance
position of a cross-peak was significantly shifted compared to the reference spectrum. The active
compound was obtained through successive deconvolution. Experiments were recorded on a Bruker
DRX600 spectrometer equipped with a triple-resonance cryoprobe. The preparation of samples was
done automatically by a Tecan Genesis RSP 150 pipetting robot. Spectra were analysed using the
programs TOPSPIN and SPARKY.[47]

**Synthesis of compounds**
All reagents and starting materials were purchased from Sigma-Aldrich Chemie GmbH, ABCR GmbH
& Co.KG, alfa Aesar GmbH & Co.KG or Acros Organics and used without further purification. All air
or moisture-sensitive reactions were carried out under dry nitrogen using standard Schlenk techniques.
Solvents were removed by evaporation on a Heidolph Laborota 4000 with vacuum provided by a PC
3001 Vaccubrand pump. Thin-layer chromatography (TLC) was performed on plastic-backed plates pre-



coated with silica gel 60 $F_{254}$ (0.2 mm). Visualization was achieved under an ultraviolet (UV) lamp (254
and 366 nm). Flash chromatography was performed using J.T Baker silica gel 60 (30-63 µm). Analytical
high-performance liquid chromatography (HPLC) was performed on a Shimadzu LC-20 (degasser
DGU-20A3, controller CBM-20A, autosampler SIL-20A) with a DAD-UV detector (SPD-M20A),
using a reverse-phase C18 column (Nucleodur 100-5, 5 µM, 250 mm x 4 mm, Macherey-Nagel, Düren,
Germany). Separation of compounds by preparative HPLC was performed on a Shimadzu LC-8A
system equipped with a UV detector (SPD-M20A), using a semi-preparative C18 column (Nucleodur
100-5, 5 µM, 250 mm x 10 mm, Macherey-Nagel) or preparative C18 column (Nucleodur 100-5, 5 µM,
250 mm x 21 mm, Macherey-Nagel). The detection wavelength was 254 nm. Gradients of acetonitrile-
water with 0.1% TFA were used for elution at flow rates of 1 mL/min, 8 mL/min, and 14 mL/min on
the analytical, semi-preparative and preparative columns respectively. Melting points (mp) were
determined with Stuart Melting Point Apparatus SMP3 and are not corrected. Mass spectra were
recorded on a 4000Q TRAP LC/MS/MS/ System for AB Applied Biosystems MDS SCIEX. NMR
spectra were recorded on a Bruker AV300 spectrometer instrument operating at 300 MHz for proton
frequency using DMSO-*d6* solutions. Chemical shifts were quoted relative to the residual DMSO peak
($^1$H: δ = 2.50 ppm, $^{13}$C: δ = 39.52 ppm). Coupling constants (J) are given in Hertz (Hz). Splitting patterns
are indicated as follows: singlet (s), doublet (d), triplet (t), quartet (q), multiple (m), broad (b). Purity of
each compound used for biological testing was ≥95% unless otherwise noted. The purity check of known
inhibitors purchased for comparison with our compounds are found in Supporting Information Figure
S9.

**Synthesis of compounds 8, 11 – 17**






R$_1$ = CF$_3$ , Br, Cl

R$_2$ =

**Scheme 3**: Synthesis of compounds 8, 11 - 17

To a solution of anthranilic acid substituted with the appropriate R$_1$ (1.32 mmol) and sodium carbonate

(3.17 mmol) in water (2 mL) at 80 °C, the sulfonyl chloride (1.58 mmol) substituted with the appropriate

R$_2$ was added over a period of 5 minutes. The stirring continued for 18 h at 80 °C. The reaction mixture

was cooled to room temperature and acidified with 6 N HCl, and the resulting solid precipitate was

filtered, washed with water and dried to give the crude product. The final product was obtained by

preparative HPLC (Puranik 2008).

**2-(5,6,7,8-tetrahydronaphthalene-2-sulfonamido)- 5- (trifluoromethyl) benzoic acid (8)**

(0.52 g, 74% yield) **$^1$H-NMR** (300 MHz, DMSO-d6): δ = 11.77 [s, 1H, CO*OH*], 8.13 [s, 1H, *NH*],

7.85 [d, $^3$J$_{6,4}$ = 2.1 Hz, 1H, 6-H$_{Ar}$] 7.62 [d, $^4$J$_{1',3'}$ = 2.1Hz , 1H , 1'-H$_{Ar}$] 7.53 [ dd, $^3$J$_{4,3}$ = 7.1 Hz, $^4$J$_{4,6}$ =

2.1 Hz, 4-H$_{Ar}$] 7.36 [dd, $^3$J$_{3'4'}$ = 7.5 Hz, $^4$J$_{3',1'}$ = 2.4 Hz, 1H, 3'-H$_{Ar}$] 7.15 [d, $^3$J$_{4',3'}$ = 7.5Hz, 1H,4'-H$_{Ar}$ ]

, 6.90 [d, $^3$J$_{3,4}$ = 7.1Hz, 1H , 3-H$_{Ar}$] 2.73 (m, 4H, CH$_2$); 1.6 ( m, 4H, CH$_2$);-**$^{13}$C-NMR** (75 MHz, DMSO-

d6): δ = 169.1(C, C$_{Ar}$-8], 152.7(C, C$_{Ar}$-2), 143.8 (C, C$_{Ar}$-4'), 138.7(C, C$_{Ar}$-2'), 135.9 (C, C$_{Ar}$-8a'),

130.4(CH, C$_{Ar}$-4), 128.7 (CH, C$_{Ar}$-6), 127.5 (CH, C$_{Ar}$-1'), 124.0 (CH, C$_{Ar}$-4'), 121.6 (C, C-6), 118.2 (C,

C$_{Ar}$-5), 116.9 (C, C$_{Ar}$-3), 29.0 (CH$_2$, C-8'), 28.8 (CH$_2$, C-5'), 22.3 (CH$_2$, C-6'), 22.2 (CH$_2$, C-7'); mp:

177°C; MS (ESI) *m*/z:calcd. for C$_{18}$H$_{16}$F$_3$NO$_4$S, 399; found 400 [M+H]$^+$.

**5-bromo-2-(naphthalene-2-sulfonamido) benzoic acid (11)**

(0.13 g, 67% yield) **$^1$H-NMR** (300 MHz, DMSO-d6): δ = 10.2 [s, 1H, *COOH*], 9.8 [s, 1H, *NH*] 8.59 [d,

$^4$J$_{1',3'}$ = 1.4 Hz, 1 H, 1'-H$_{Ar}$], 8.17 [d, $^3$J$_{8',7'}$ = 7.8 Hz, 1 H, 8'-H$_{Ar}$], 8.10 [d, $^3$J$_{4'3'}$ = 8.8 Hz, 1 H, 4'-H$_{Ar}$],

8.02 [d, $^3$J$_{5',6'}$ = 7.8 Hz, 1 H, 5'-H$_{Ar}$], 7.93 [d, $^4$J$_{6,4}$ = 2.4 Hz, 1 H, 6-H$_{Ar}$], 7.77 [dd, $^3$J$_{3',4'}$ = 8.8 Hz, $^4$J$_{3',1'}$

= 1.4Hz ,1 H, 3'-H$_{Ar}$], 7.72 – 7.65 [m, 3 H, 4-H$_{Ar}$, 6'-H$_{Ar}$, 7'-H$_{Ar}$] , 7.51 [d, $^3$J$_{3,4}$ = 8.9 Hz, 1 H, 3-H$_{Ar}$]. −

**$^{13}$C-NMR** (75 MHz, DMSO-d6): δ = 168.2 (C, C-7) ,138.8 (C, C$_{Ar}$-2),136.8 (CH, C$_{Ar}$-4),135.3 (C, C$_{Ar}$-

4a'),134.4 (C, C$_{Ar}$-8a') ,133.4 (CH, C$_{Ar}$-6) ,131.4 (CH, C$_{Ar}$-6'), 129.3 (CH, C$_{Ar}$-4') ,128.5 (CH, C$_{Ar}$-



8′),127.8 ( 2xCH, C$_{Ar}$-5′, C$_{Ar}$-7′) 121.6 (CH, C$_{Ar}$-3′),120.6 (CH, C$_{Ar}$-3),119.0( C, C$_{Ar}$-1) ,114.9 (C, C$_{Ar}$-
5). Mp: 199°C; (ESI) m/z: calcd.for C$_{17}$H$_{11}$BrNO$_4$S$^-$ ; 403.9560: found   403.9613 [M-H]$^-$.

**5-bromo-2-(phenylmethylsulfonamido)benzoic acid (12)**

(0.07g, 42% yield) **$^1$H-NMR** (300 MHz, DMSO-d6): δ = 10.57 [ s, 1 H, COO*H*], 8.05 [d,  $^4$J$_{6,4}$ = 2.4
Hz, 1 H, 6-H$_{Ar}$], 7.75 [dd,  $^3$J$_{4,3}$ = 8.9 Hz, $^4$J$_{4,6}$ = 2.4Hz ,1 H, H-4$_{Ar}$], 7.49 [d,$^3$J$_{3,4}$ = 8.9 Hz,1 H, 3-H$_{Ar}$],
7.33 – 7.28 [m, 3 H, 3′-H$_{Ar}$ , 5′-H$_{Ar}$], 7.23 – 7.20 [m, 2 H, 4′-H$_{Ar}$], 5.75 [s, 1 H, N*H*], 4.72 [s, 2 H, 1′-
H] $^{13}$**C-NMR** (75 MHz, DMSO-d6): δ = 168.3 (C, C-7) , 139.9 (C, C$_{Ar}$-2), 137(CH, C$_{Ar}$-4), 133.4 (CH,
C$_{Ar}$-6), 130.7 (CH, C$_{Ar}$-3′) , 128.6 ( C, C$_{Ar}$-2′) , 128.4 (CH, C$_{Ar}$-5′) , 128.3 (CH, C$_{Ar}$-4′) , 119.5 (CH,
C$_{Ar}$-3) , 117.5 (C, C$_{Ar}$-1)   , 113.9 (C, C$_{Ar}$-5) , 57.4 (CH$_2$, C-1′). Mp: 216°C; (ESI) m/z: calcd.for
C$_{14}$H$_{11}$BrNO$_4$S$^-$ 367.9860; found 367.9878 [M-H]$^-$.

**5-bromo-2-(4-(phenoxymethyl)phenylsulfonamido)benzoic acid (13)**

(0.6 g, 29% yield) 1H-NMR (300 MHz, DMSO-d6): δ = 7.97 [d, $^4$J$_{6,4}$ = 2.4 Hz, 1 H, 6-H$_{Ar}$), 7.85 (d,
$^3$J$_{2′,3′}$ = 8.3 Hz, 2 H, 3′-H$_{Ar}$), 7.73 [dd, $^3$J$_{4,3}$ = 8.9 Hz, $^4$J$_{4,6}$ = 2.4Hz , 1 H,4-H$_{Ar}$], 7.63 [d, $^3$J$_{2′,3}$′ = 8.3 Hz, 2
H, 2′-H$_{Ar}$], 7.47 [d, $^3$J$_{3,4}$ = 8.9 Hz, 1 H, 3-H$_{Ar}$], 7.29 [dd, $^3$J$_{3″,2″}$ = $^3$J$_{3″,4″}$=7.3 Hz,  2 H, 3″-H$_{Ar}$], 7.00 – 6.92
[m, 3 H, 4″-H$_{Ar}$ , 2″-H$_{Ar}$], 5.17 [s, H, 5′-H]. − **13C-NMR** (75 MHz, DMSO-d6): δ = 168.2 (C, C-7) ,
157.9 ( C, C$_{Ar}$-1″) , 143.2 (C, C$_{Ar}$-4′), 138.8 (C, C$_{Ar}$-2), 137.5 (C, C$_{Ar}$-1′) , 136.9 (CH, C$_{Ar}$-4) 133.5 (CH,
C$_{Ar}$-6), 129.4(CH, C$_{Ar}$-3″), 128.1(CH, C$_{Ar}$-2′),127.0 (CH, C$_{Ar}$-3′) , 120.9 (CH, C$_{Ar}$-4″) , 120.5 (CH, C$_{Ar}$-
3) , 119.0 (C, C$_{Ar}$-1) , 114.9(CH, C$_{Ar}$-5), 114.7 (CH, C$_{Ar}$-2″), 68.0 (CH$_2$, C-5′) Mp: 175°C; (ESI)  m/z:
calcd for C20H15BrNO5S- 459.9860 found 459.9878 [M-H]-.

**5-bromo-2-(2,4,6-trimethylphenylsulfoamido)benzoic acid (14)**

(0.6 g, 78% yield) **$^1$H-NMR** (300 MHz, DMSO-d$_6$): δ = 11.77 [s,1H, *COOH*], 9.98 [s, 1H, *NH*], 7.68
[d, $^3$J$_{6,4}$ = 7.4 Hz, 1H, 6-H$_{Ar}$], 7.51[dd, $^3$J$_{4,3}$ =7.1Hz,  $^4$J$_{4,6}$ =7.4 Hz,  1H 4-H$_{Ar}$], 7.17 [d, 2H, 4′-H$_{Ar}$ , 6′-
H$_{Ar}$],  7.14 [d, $^3$J$_{3,4}$= 1H,3-H$_{Ar}$] , 2.56 [s, 6H, CH$_3$, 9′-H, 7′-H],  2.21 [s, 3H, CH$_3$, 8′-H];-$^{13}$**C-NMR** (300
MHz, DMSO-d$_6$): δ = 168.8 (C, C-7), 143.3 (C, C$_{Ar}$-2), 139.5 (C, C$_{Ar}$-2′), 139.0 and 139.0 ( 2xC, C$_{Ar}$-
3′, C$_{Ar}$-1′) 137.3 (CH, C$_{Ar}$-4), 134.0 (CH, C$_{Ar}$-6′), 133.0 (CH, C$_{Ar}$-6), 132.5  and 132.5 ( 2XCH, C$_{Ar}$-4′,
C$_{Ar}$-6′) 119.1(CH, C$_{Ar}$-3), 117.9(C, C$_{Ar}$-5), 114.3 (C, C$_{Ar}$-1), 22.5 and 22.5 (2 x CH$_3$, C-7′, C-9′)  20.7
(CH$_3$, C-8′) ;   mp: 185; MS (ESI): m/z 399 [M+H]+.

**2-(4-acetylphenylsulfoamido)-5-(trifluoromethyl)benzoic acid (15)**



(0.4 g, 63% yield) **$^1$H-NMR** (300 MHz, DMSO-d6): δ = 12.28 [s, 1H, *COOH*]; 12.10 [s, 1H, *NH*], 8.11
[d, $^4J_{6,4}$= 2.5 Hz, 1H, 6-H$_{Ar}$], 8.08 [d, $^3J_{3'2'}$ = 7.5 Hz, 2H, 3'-H$_{Ar}$], 7.86 [dd, $^4J_{4,6}$= 2.5 Hz, $^3J_{4,3}$ = 7.3Hz,
1H, 4-H$_{Ar}$], 7.64 [d, $^3J_{4,3}$ = 7.3 Hz, 1H, 3-H$_{Ar}$] , 7.56 [dd $^3J_{2',3'}$ = 7.5Hz,   $^4J_{2',6'}$ =2.3Hz,  2H, 2'-H$_{Ar}$, 6'-
H$_{Ar}$ ]  7.22 [dd , $^3J_{3',2'}$ = 7.5Hz , $^4J_{3',5'}$ = 2.1Hz , 2H, 3'-H$_{Ar}$, 5'-H$_{Ar}$]  2.50 [s, 3H, CH$_3$, 8'-H]; - **$^{13}$C-NMR**
(75 MHz, DMSO-d6): δ = 197.9 (C, C-7'), 169.1(C, C-8), 151.8 (C, C$_{Ar}$-2) 143.5 (C, C$_{Ar}$-1'), 142.5 ( C,
C$_{Ar}$-4'), 140.6 (CH, C$_{Ar}$-4), 131.4 (CH, C$_{Ar}$-7), 129.6 (2XCH, C$_{Ar}$-3', C$_{Ar}$-5'), 128.6 (2xCH, C$_{Ar}$-2', C$_{Ar}$-
6'), 127.6 (C, C$_{Ar}$-6),  123.0 (C, C-$_{Ar}$-5), 118.7 (CH, C$_{Ar}$-3), 27.3 (CH$_3$, C-8'); mp: 170°C; MS (ESI)  *m*/z
: calcd. for C$_{16}$H$_{12}$F$_3$NO$_5$S. 387; found 388 [M+H]$^+$.
**2-(2,3-dihydrobenzo[*b*][1,4]dioxine-6-sulfonamido)-5-(trifluoromethyl)benzoic acid (16)**
(0.4 g, 65% yield) **$^1$H-NMR** (300 MHz, DMSO-d6): δ = 11.48 [s, 1H, *COOH*], 8.13[s, 1H, *NH*] , 7.89
[d, $^4J_{6,4}$ = 3,9 Hz , 1H , 6-H$_{Ar}$]  7.66 [dd, $^3J_{4,3}$ = 7.2 Hz, $^4J_{4,6}$ = 4.3 Hz, 1H, 4-H$_{Ar}$],
7.23 [d, $^3J_{4,3}$ = 7,2 Hz 1H, 3-H$_{Ar}$],   7.11 [dd, $^3J_{2',3'}$ = 7.3Hz , $^4J_{2',8'}$= 3.2Hz , 1H, 2'-HAr ] 6.95 [ d, $^4J_{2',8'}$=
3.2 Hz, 1H , 8'-H$_{Ar}$ ]  4.23 – 4.31 [m, 4H, 5'-H, 6'-H ]; - **$^{13}$C-NMR** (75-MHz, DMSO-d6): δ = 168.9(C,
C-8), 148.3( C, C-4'), 143.8 (C, C-2), 143.5 (C, C-7') , 131.3 (C, C-1'), 130.8 (CH, C-4), 128.6(CH, C-
6),  125.7 (C, C-7), 122.1( C, C-5), 120.9(CH, C-2'), 118.3 (CH, C-3), 118.1(CH, C-3'), 116.8 (CH, C-
8') , 64.7(CH$_2$, C-5') 64.3 ( CH$_2$, C-6'); mp: 178°C;  MS (ESI) *m*/z: calcd. for C$_{16}$H$_{12}$F$_3$NO$_6$S. 403; found
[M+H]$^+$.
**5-(trifluoromethyl)-2-(2,4,6-trimethylphenylsulfoamido)benzoic acid (17)**
(0.38 g, 62% yield) **$^1$H-NMR** (300 MHz, DMSO-d6): δ = 12.28 [s, 1H, *COOH*], 11.60 [s, 1H, *NH*], 8.15
[d, $^4J_{6,4}$ = 4.3 Hz, 1H, 6-H$_{Ar}$] 7.92 [dd, $^3J_{4,3}$ =7.9 Hz, $^4J_{4,6}$ = 2.1 Hz, 1H,4-H$_{Ar}$] 7.87 [d, $^4J_{6',4'}$= 1.9 Hz, 2H,
4'-H$_{Ar}$ , 6'-H$_{Ar}$],  7.48 [d, $^3J_{3,4}$ = 7.9 Hz, 1H, 3-H$_{Ar}$] , 2.60 [s, 6H, CH$_3$ , 9'-H, 7'-H], 2.23 [s, 3H, CH$_3$,
8'-H]; - **$^{13}$C-NMR** (75 MHz, DMSO-d6): δ = 169.3 (C, C-7), 154.2 (C, C-2), 143.6 (C, C-2') , 139.1
and 139.1 (2xC, C-1', C-3') 132.9 (C, C-5'),  132.5 (CH, C-4), 131.5 and 131.5 (2xCH, C-4', C-6') ,
130.1(CH, C-6), 128.7 (C, C-8), 122.5 ( C, C-5), 117.0 (CH, C-3), 109.0 (C, C-1) ,22.4 and 22.4
(2xCH$_3$,C-7', C-9'), 20.8 (CH$_3$, C-8') ; mp:184°C; MS (ESI) *m*/z: calcd. for C$_{17}$H$_{16}$F$_3$NO$_4$S; 387: found
388 [M+H]$^+$.



**18, 19, 20,** and **21** were purchased from Enamine, Kiev, Ukraine as pure compounds (see also Table S6,
Supporting Information).

**Determination of ligand binding and binding constant by NMR**
50 µM of $^{15}$N-labeled protein samples were prepared in a 20 mM sodium phosphate buffer containing
50 mM sodium chloride, 0.02 % (w/v) NaN$_3$, at pH 7.4. Stock solutions of small molecules were
prepared in DMSO-*d6* at a concentration of 160 mM. A $^1$H-$^{15}$N HSQC spectrum of Dvl PDZ was
acquired at 300 K with 5% DMSO-*d6* in the absence of ligand as reference spectrum. Mixtures of 16
compounds were added to $^{15}$N-labeled Dvl PDZ at 8-fold molar excess each. The final concentration of
DMSO-*d6* in the protein-ligand solutions was 5%. Spectra were acquired with 8 scans and 256 points
in the indirect dimension.
Binding was deduced if the resonance position of a cross-peak was significantly shifted compared to the
reference spectrum. The active compound was obtained through successive deconvolution. Experiments
were recorded on a Bruker DRX600 spectrometer equipped with a triple-resonance cryoprobe. The
preparation of samples was done automatically by a Tecan Genesis RSP 150 pipetting robot. Spectra
were analysed using the programs TOPSPIN and SPARKY.
Chemical shift perturbations were obtained by comparing the $^1$H-$^{15}$N backbone resonances of protein
alone to those of protein-ligand complex. The mean shift difference ($\Delta\delta$ in ppm) was calculated using
the equation 1 (Garrett 1997, Bertini 2011).

$$\Delta\delta = \sqrt{\left[\frac{1}{2}(\Delta\delta_H)^2 + \frac{1}{25}(\Delta\delta_N)^2\right]} \quad (Eq.\,1)$$

Here $\Delta\delta_N$ and $\Delta\delta_H$ are the amide nitrogen and amide proton chemical shift differences between the free
and the bound states of the protein. In order to estimate binding constants, titration experiments
monitored by NMR were done. A series of $^1$H-$^{15}$N HSQC were recorded as a function of ligand
concentration. Residues showing a continuous chemical shift change and for which the intensity
remained strong were classified as being in fast exchange. The dissociation binding constant was
estimated by fitting the observed chemical shift change to equation 2 (Shuker 1996, Hajduk 1997).

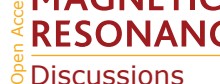

$$\frac{\Delta\delta}{\Delta\delta_{max}}$$
$$= \frac{([L_T] + [P_T] + K_D) - \sqrt{([L_T] + [P_T] + K_D)^2 - 4[L_T]\cdot[P_T]}}{2[P_T]} \quad (Eq.2)$$

$\Delta\delta$ is the observed protein amide chemical shift change at a given compound concentration and $\Delta\delta_{max}$
the maximum chemical shift change at saturation. $[L_T]$ the total concentration of the compound, and $[P_T]$
the total concentration of the protein. $K_D$ is the equilibrium dissociation constant. The $K_D$ values are
reported as means ± standard deviations of at least six residues influenced upon binding of the ligand.

**Determination of binding constant by Isothermal Titration Calorimetry (ITC)**
Isothermal Titration Calorimetry (ITC) experiments were performed using a VP-ITC system
(MicroCal). Protein in 20 mM Hepes buffer, 50 mM NaCl, pH 7.4, was centrifuged and degassed before
the experiment. A 200 μM ligand solution containing 2% DMSO was injected 30 times in 10 μL aliquots
at 120 s intervals with a stirring speed of 1000 rpm into a 1.4 mL sample cell containing the Dvl PDZ
domain at a concentration of 20 μM at 25 °C. Control experiment was initially determined by titrating
ligand into buffer at same conditions. Titration of ligand into buffer yielded negligible heats.
Thermodynamic properties and binding constants were determined by fitting the data with a nonlinear
least-squares routine using a single-site binding model with Origin for ITC v.7.2 (Microcal).

**Protein expression**
PDZ domains of human AF6 (P55196-2, residues 985–1086) and murine α1-syntrophin (Q61234,
residues 81–164) were cloned into pGEX-6P-2 (Amersham Biosciences, Freiburg, Germany) and
pGAT2 (European Molecular Biology Laboratory, Heidelberg, Germany), respectively. Proteins were
expressed in *E. coli* BL21 (DE3) cells and purified as previously described (Boisguerin 2004). For the
cloning of the Dvl-1 PDZ domain (O14640, residues 245–338), IMAGp958J151157Q (ImaGenes) was
used as template. V250 is exchanged to isoleucine as in human Dvl-3 or murine Dvl-1. The C-terminal
C338 of the domain was exchanged by serine. Via cloning in pET46EK/LIC, a coding sequence for a
TEV (Tobacco Etch Virus) protease cleavage site was introduced. The resulting plasmid pDVL1 was



transformed in *E. coli* BL21 (DE3). Expression on two-fold M9 minimal medium with 0.5 g/L $^{15}$N
NH$_4$Cl as sole nitrogen source in shaking culture was done at 25 °C overnight with 1 mM IPTG. A yield
of 25 mg of pure Dvl-1 was obtained from 1 L culture after IMAC, TEV protease cleavage, a second
IMAC, and gel filtration (Superdex 75). The protein domain Dvl-1_245–338 was supplied for NMR in
20 mM phosphate buffer, pH 7.4, 50 mM NaCl.
The production of Dvl-3 (Q92997 residues 243-336), mShank3 (Q4ACU6, residues 637-744) PDZ
domains and the 3 PDZ domains of PSD95 was described by Saupe et al (Saupe 2011).

**Crystallization and X-ray diffraction**

The His-tagged cleaved human Dvl-3 PDZ domain was concentrated to 12-20 mg/mL in the presence
of a 5-fold molar excess of compound **3**, **5**, **6**, **7**, **11** and **12**. Crystals of all complexes were grown at
room temperature by the sitting drop vapour-diffusion method. 200 nL Dvl-3/compound solution was
mixed with an equal volume of reservoir solution using the Gryphon (Formulatrix) pipetting robot.
Crystals of all complexes were grown to their final size within 4 to 14 days. The Dvl-3 PDZ domain
crystallized in complex with compound **3** and **7** in crystallization condition 30% PEG 8000, 0.2 M
ammonium sulphate, 0.1 M MES pH 6.5; with compound **5** in 30% PEG 8000, 0.1 M MES pH 5.5; with
compound **6** in 1.2 M ammonium sulphate, 0.1 M citric acid pH 5.0; with compound **12** in 32% PEG
8000, 0.2 M ammonium sulphate, 0.1 M Na-cacodylate pH 6.0; with compound **11** in 1 M ammonium
sulphate, 1% PEG 3350, 0.1 M Bis-Tris pH 5.5; with compound **12** in 1.26 M sodium phosphate, 0.14
M potassium phosphate and with compound **18** in 1.5 M ammonium sulphate, 12% glycerol, 0.1 M Tris-
HCl pH 8.5. The crystals were cryoprotected if necessary, for data collection by soaking for few seconds
in precipitant solution containing 20% (v/v) glycerol and subsequently frozen in liquid nitrogen.
Diffraction data were collected at 100 K at beamline BL14.1 at the synchrotron-radiation source BESSY,
Helmholtz-Zentrum Berlin and processed with XDS.

**Structure determination and refinement**

Phases for the Dvl-3 PDZ domain in complex with compound 3 were obtained by molecular replacement
with PHASER (McCoy 2007) using the *Xenopus laevis* Dishevelled PDZ domain structure (PDB code

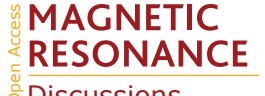

2F0A) as a starting model. The reasonable crystal packing and electron density allowed further model
and compound building using the program COOT (Emsley 2004) with iterative refinement with
REFMAC (Murshudov 1997). All further complex structures were obtained in the same way but using
the final refined compound free Dvl-3-PDZ structure as model for molecular replacement. The
Ramachandran statistics were analysed by Molprobity (Chen 2010) for all complexes and all
crystallographic statistics are given in Supporting Information Tables S2 and S3. Figures were prepared
with PyMol. Atomic coordinates and structure factor amplitudes for DVL-3 PDZ domain in complex
with compound **3**, **5**, **6**, **7**, **11**, **12**, **18** were deposited in the Protein Data Bank with accession codes
6ZBQ, 6ZBZ, 6ZC3, 6ZC4, 6ZC6, 6ZC7 and 6ZC8, respectively.

**MTT assay**
HEK293 cells were plated on a 96-well plate and treated with different concentrations of Dvl inhibitors.
After 24 h treatment, 20 μl of MTT solution (5 mg/mL) was added into each well. After 2 h incubation,
cell culture medium was replaced with 50 μL DMSO, and the signal of the purple formazan, produced
by living cells, was measured by a plate reader.

**TOP-GFP reporter assay**
The lentivirus particle (CCS-018L, SABiosciences) encoding GFP under the control of a basal promoter
element (TATA box) joined to tandem repeats of a consensus TCF/LEF binding site was transfected
into HEK293 cells. Stable cells were selected by puromycin (2 μg/mL) treatment. Wnt signalling
activity indicated by GFP intensity was measured by flow cytometry after 24 h incubation with
recombinant mouse Wnt3a (100 ng/mL) or GSK3 inhibitor CHIR99021 (3 μM) in the presence of Dvl
inhibitors.

**Luciferase reporter assays**
Plasmids encoding a firefly luciferase reporter gene under the control of different responsive elements
were transfected into Hela cells with a pRL-SV40 normalization reporter plasmid using the
Lipofectamine 2000 (Invitrogen). After desired treatment, cells were harvested in the passive lysis buffer



(Promega), and 15 µL cell lysate were transferred to 96-well LumiNunc plates (Thermo Scientific).
Firefly luciferase and Renilla luciferase were detected with the D-luciferin buffer (75 mM Hepes, 4 mM
$MgSO_4$, 20 mM DTT, 100 µM EDTA, 0.5 mM ATP, 135 µM Coenzyme A and 100 µM D-Luciferin
sodium salt, pH 8.0) and the coelenterazine buffer (15 mM Na4PPi, 7.5 mM NaAc, 10 mM CDTA, 400
mM $Na_2SO_4$, 25 µM APMBT and 1.1 µM coelenterazine, pH 5.0) respectively using the CentroXS
LB960 lumiometer (Berthold Technologies).

**Immunoblotting**
To assess the β-catenin accumulation in Hela cells, cells were treated with Wnt3a in the presence of Dvl
inhibitors for 24 h and lysed in RIPA buffer (50 mM Tris, pH 8.0, 1% NP-40, 0.5% deoxycholate, 0.1%
SDS, 150 mM NaCl). Equal amounts of protein were loaded on a SDS-PAGE. Separated proteins were
blotted onto PVDF membranes for immunoblot analysis using anti-β-catenin antibody (610154, BD).
HRP-conjugated anti-mouse antibody (715-035-150, Jackson ImmunoResearch laboratories) was used
for secondary detection with Western lightning chemiluminescence reagent plus (PerkinElmer) and
Vilber Lourmat imaging system SL-3.

**qRT-PCR analysis**
To measure the Wnt target accumulation at mRNA level, Hela cells were treated with Wnt3a in the
presence of Dvl inhibitors for 24 h. mRNA was extracted according to the standard TRIzol® protocol
(Invitrogen) and reverse-transcribed using random primers (Invitrogen) and M-MLV reverse
transcriptase (Promega). The qRT-PCR was performed in iQ5 Multicolor Real-Time PCR Detection
System (Bio-Rad) using SYBR® Green (Thermo Scientific) and gene-specific primer pairs of Bmp2,
Axin2, Lef1 and β-actin (endogenous control).

**Migration assay**
Cell motility was assessed using 24-well transwell (pore diameter: 8 µm, Corning). SW480WL cells
were seeded in the upper chamber in serum free DMEM with 0.1% BSA; 20% serum was supplemented
to medium in the lower chamber. After incubation with Wnt3a in the presence of Dvl inhibitors for 24

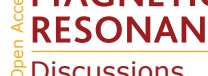

h, nonmigrant cells were scraped off using a cotton swab; the migrated cells on the filters were stained
with DAPI, photographed and counted.

**Colon sphere culture**
SW480WL cells were trypsinised into single cells, seeded on 24-well cell culture plates precoated with
250 µl polyhema (12 mg/mL in 95% ethanol, Sigma) per well, and incubated with Wnt3a in the presence
of Dvl inhibitors in the sphere culture medium (F12 : DMEM 1 : 1, 1X B-27 supplement, 20 ng/mL
EGF, 20 ng/mL FGF, 0.5% methylcellulose) for 10 days. Numbers of spheres were then counted under
the microscope.

**Notes**
The authors declare no competing financial interest.

**ACKNOWLEDGEMENTS**
This research was supported by the Deutsche Forschungsgemeinschaft (DFG) Research Group 806 and
the EU-project iNext (Infrastructure for NMR, EM and X-rays for Translational Research, GA 653706).
We thank M. Leidert and S. Radetzki for protein preparation, and B. Schlegel for NMR assistance. We
also thank E. Specker for compound analysis.
**ABBREVIATIONS USED**
NMR, nuclear magnetic resonance; HSQC, Heteronuclear Single Quantum Correlation;
AU, asymmetric unit; SAR, derive structure activity relationships; vdW, van der Waals;
ITC, Isothermal titration calorimetry; PDZ, PSD95/Disc large/Zonula occludens 1);
Dvl, Dishevelled; PPI, protein-protein interactions; PDB, Protein Data Bank;
CSP, chemical shift perturbation; GFP, green fluorescent protein; DMSO, dimethyl sulfoxide;
PEG, polyethylene glycol; RNA, ribonucleic acid; mRNA, messenger RNA;
qRT-PCR, quantitative real-time polymerase chain reaction; DMEM Dulbecco's modified Eagle's
medium ; BSA, bovine serum albumin;



**ASSOCIATED CONTENT**

**Accession Codes**
Atomic coordinates and structure factor amplitudes for DVL3 PDZ domain in complex with compound
**3**, **5**, **6**, **7**, **11**, **12**, **18** were deposited in the Protein Data Bank with accession codes 6ZBQ, 6ZBZ, 6ZC3,
6ZC4, 6ZC6, 6ZC7 and 6ZC8, respectively. Authors will release the atomic coordinates and
experimental data upon article publication.

**Supporting information**
**1.** Structure-based alignment of the amino acid sequences of Dvl-1,2,3 PDZ ; PSD95-PDZ-1,2,3 ;  Af-
6 and Syn PDZ domains. (S.2)
**2.** 1H-15N HSQC spectra of Dvl-3 PDZ domain alone and in the presence of varying concentrations of
compound 3. (S.3)
**3.** Detailed views of diverse compounds bound to the Dvl-3 PDZ domain. (S.4)
**4.** Cell viability assays of compounds 3, 7,8, 9, 10, (A) and 18, 20, 21 (B). (S.5)
**5.** ITC binding assays of compound 18 with Dvl-3 PDZ (A) and with Dvl-1 PDZ (B). (S.5)
**6.** Structures of selected compounds used for comparison to our compounds. (S.6)
**7.** ITC data of selected compounds used for comparison to our compounds. (S.7)
**8.** Definition of PDZ binding site. (S.8)
**9.** Purity check of compounds. (S.9)

– Purity check of NPL-1011 compound. (S.9)

– Purity check of Sulindac compound. (S.10)

– Purity check of CalBioChem-322338 compound. (S.11)

– Purity check of NSC668036 compound. (S.12)

– LCMS of intermediate compound 8. (S.13)

– LCMS of intermediate compound 14. (S.13)

**10.** Chemical shift perturbation values of Dvl-3 PDZ and Dvl-1 PDZ for compounds (3-21). (S.14)
**11.** Data collection and refinement statistics of compounds 3, 5, 6, 7. (S.15)



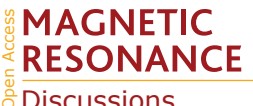

**12.** Data collection and refinement statistics of compounds 11, 12, 18. (S.16)
**13.** Selectivity of ligands derived from chemical shift perturbation of compounds tested at other PDZ
domains. (S.17)
**14.** Details of Multifilter routines. (S.17)
**15.** Smiles codes and Compounds ID. (S.18)
**16.** NMR characterization of synthesized compounds (8, 11, 13, 14, 15, 16, 17). (S.21)

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
