# Peer review of "Small-molecule inhibitors of the PDZ domain of Dishevelled proteins interrupt Wnt signalling"

_Magnetic Resonance, 2021_

## Author Response (AR1)

FMP | ROBERT-RÖSSLE-STR. 10, 13125 BERLIN

**The Editor, Magnetic Resonance**

Prof. Dr. Hartmut Oschkinat

Campus Berlin-Buch Robert-Rössle-Str. 10 13125 Berlin www.leibniz-fmp.de

oschkinat@fmp-berlin.de

Phone: +49 30 94793 160

Berlin, 26.04.2020

**Dear Editor,**

Herewith we submit the revised version of our manuscript 'Small-molecule inhibitors of the PDZ domain of Dishevelled proteins interrupt Wnt signalling' by Kamdem et al. We have responded to all criticisms and suggestions of the referees and in the community comment. In particular, we have measured binding of the compound CBC-322338/3289-8625 to DVL-3 PDZ by NMR as a backup of the ITC experiments. Based on the observed chemical shift changes it binds less than 50 micromolar, according to a comparison between the chemical shift changes presented in Table S1 and the KD in Table 1. However, we have also put our comments a little more into perspective by explaining that the diverging results were obtained using different assays. Of course, we have explained in all relevant Figure legends how errors were calculated, how many data points were taken, and how many molecules were binding (n-value in ITC).

Our point-by-point reply cites the referee first (we did not copy in the sections with positive comments), without copying the whole statement into this letter, then our comment, then the new text in the manuscript.

**Referee 1, Mingjie Zhang**

Thank you very much for the encouraging comments.

Minor concerns:

- 1. There appear discrepancies in the bindings of compound 3289-8625 to Dvl PDZ in the literature (Grandy et al, 2009) and in the current study based on the data in Fig. S7. The assay condition used in the current study may not allow the authors to derive accurate binding constant of the interaction. The authors need to address this issue.
- 2. In line 185&187: "R2" should be "R1".

This referee indicated that the results associated with compound 3289-8625 should be discussed

differently. We did so, and please have also a look at the reply to Jie Zheng, his community comment, were we discussed the NMR shift assay now performed for this revision (see new Figure S8), in more detail. In reply to this request and the other, we changed/included the sentence

'We also applied an NMR shift assay (Figure S8), yielding a  $\triangle$ CSP around 0.1. Based on NMR and ITC studies, the binding affinity of CBC-322338/3289-8625 to DVL-3 seems to be less than 50 micromolar (comparing the CSPs from the NMR assay with those of our other compounds listed in Table S1 and the respective binding constants in Table 1, and considering also the weak heat development in our ITC assay) which was larger than the originally reported value (10.6 +/- 1.7) (Grandy 2009) that was obtained with a different method.' (lines 359-366)

Of course, we replaced R2 by R1 in former lines 185 and 187, now 272 and 274.

**Referee 2 (Anonymous)**

**Introduction:** Perhaps the authors can rearrange some paragraphs to simplify the background description for the non-specialist reader. It could start with the description of Dishevelled proteins (line 56) and their modular composition (please cite some references describing the DIX and the DEP domain structures). Then, it can describe the PDZ structures, their binding properties and why they became targets for drug design to treat several diseases. The authors could also mention a couple of recent reviews describing the advances in the design of peptides and small molecule modulators of PDZ domains and the logic behind this new work.

We changed the introduction as requested and included more literature regarding the design of PDZ domain inhibitors, including peptidomimetics, in particular by including a very nice review from the Strømgaard group:

'Due to their occurrence in important proteins, PDZ domains received early attention as drug targets, nicely summarized in Christensen 2019. There are several examples of DvI PDZ inhibitors of peptide or peptidomimetic nature (eg. Hammond 2006, Haugaard-Kedstrom 2021), including peptide conjugates (eg. Qin 2021, Hegedus 2021), and on an organic, small-molecule basis.' (lines 73-76)

We also discussed the scope of our work more explicitly in light of the existing data:

'On this path, we explore optimal fits for the primary binding pocket by cycles of chemical synthesis and X-ray crystallography and further avenues for systematically growing ligands along the DvI PDZ surface to provide SAR for the development of inhibitors in the low or medium nanomolar range.' (lines 89-92)

**Results:** The definition of the PDZ binding site is shown as Figure S8. The authors could consider moving this figure to the main figures (New Figure 1 panel A) and label the red and blue spheres with Dishevelled residues. Having this representation next to the complexes will facilitate the description of the binding cavity in the structures (New Figure 1 remaining panels).

We changed Figure 1 as requested by transferring the back bone cartoon shown in Figure S8 into Figure 1A, citing it several times in the Introduction as well as in the virtual screening, NMR and later sections (lines 56, 112, 135, 138, 350).

**Data presentation**: The authors should include an overlay of the corresponding data points (as dot plots) in Figure 2 (Bar charts), Figure S4 and Figure S5, and please, add the n number and define the error bars (e.g. SD, SEM) in the figure legends.

We improved the data representation by including of course the definition of the error bars and the n numbers in figure legends of S4, S5, and S7. Since the other referee was ok with the presentation of the data, suggesting even that the paper is actually quite well written, we would like to leave the bar charts in Fig. 2, since we think they represent the data quite clearly. Of course, we now indicated the number of replicas and how errors were calculated. Apart from changing Figure 1, and mentioning 1A several times, we changed the following legends by adding the following sentences:

**Table 1:**

'The  $K_D$  values determined by NMR are reported as means  $\pm$  standard deviations of measurements evalutating shifts of cross peaks of at least six residues influenced upon binding of the ligand. The  $K_D$  values (1/KA) determined by ITC were obtained as fits to a one-site binding model (n in the range of 0.95-1.2) with  $K_D$  errors obtained by  $\Delta K_A/K_A^2$ .'

**Figure 2:**

'For all tests, three independent biological replicas were performed and error bars represent standard deviations. P-values were calculated from T-test. \*: P < 0.05; \*\*: P < 0.01; \*\*\*: P < 0.001.

**Figure S4:**

'Three independent biological replicas were performed in each case and error bars represent standard deviations.'

**Figure S5:**

'The data in A and B fitted to a one-site binding model with  $K_D$  determined by  $1/K_A$  and  $\Delta K_D = \Delta K_A/K_A^2$  and with n=1.14 and 1.12, respectively.'

**Figure S7:**

'C) Compound CBC-322338/3289-862516e and D) NSC66803616a did not show any binding to the DVL3-PDZ domain under the assay conditions applied. The given KD were determined by 1/KA and the associated error by  $\Delta KD = \Delta KA/KA2$ .'

Figure S1: Please use monospaced fonts to ensure that the sequences are aligned.

Figure S1 is changed to monospaced fonts (courier, as it was before), and was checked before submission whether the format is kept.

Equation1 Please correct this equation.

**Community comment, Jie Zheng**

Most importantly, Ji Zheng requested an NMR binding assay. We have done the requested NMR binding assay with DvI-3, which indicated binding weaker than 50 micromolar. Employing an eight-fold excess of ligand (as common in this study), we observed relatively small chemical shift changes, with a maximum change of around 0.1 ppm for only one signal of the three critical ones. Table S1 lists similar chemical shift changes (but above 0.1 ppm as an average over three signals) for ligands that bind in the three-digit micromolar range (eg. compound 2 and 6). Those three shifting signals are indicated in Fig. S2, and comparison of S2 and S8 displays a considerable difference. In S2, the highest ligand concentration was 8-fold excess. Our best binding compounds show shift changes larger 0.3 ppm as an average over three signals. The compound was checked by mass spectrometry. We included the NMR binding assay employing 8-fold excess of the compound into Figure S8 and reworded the discussion related to this compound, see reply to Minye Zhang:

'We also applied an NMR shift assay (Figure S8), yielding a  $\triangle$ CSP around 0.1. Based on NMR and ITC studies, the binding affinity of CBC-322338/3289-8625 to DVL-3 seems to be less than 50 micromolar (comparing the CSPs from the NMR assay with those of our other compounds listed in Table S1 and the respective binding constants in Table 1, and considering also the weak heat development in our ITC assay) which was larger than the originally reported value (10.6 +/- 1.7) (Grandy 2009) that was obtained with a different method.' (lines 359-366)

We would like to mention that the ITC and NMR assays above were performed by different persons in different labs here on the campus in Berlin Buch, and the compound was obtained from the EU-OPENSCREEN (an EU facility) and FMP compound library that is handled under professional conditions. The compound was obtained from the facility already dissolved in DMSO and this solution was quality-controlled by LC-MS. Independently, Hori et al. found a similar binding characteristic, also in a direct binding assay.

With respect to the inclusion of the ITC and NMR results into Table 1 we made sure that the values for DvI-1 and DvI-3 are determined with the same method. We would like to mention that we do not have both values in all of the cases, especially as many values are associated with compounds synthesized at intermediate stages of the project, and one value was sufficient for finding the right path. It was important to us to provide values determined by ITC for the important compounds **18-21** since it may be considered the superior method in such investigations. However, we have reported chemical shift changes ( $\Delta$ CSP) for most of the compounds in the supplementary information (Table S1 on page 14).

We hope that manuscript is now in shape for publication and look forward to your answer, Yours sincerely,

Hartmut Oschkinat

---

## Author Response (AR2)

**Point-by-point response**

1. p.2 l.52 (Doyle 1996), Lee 2017), a ) too much – **1 bracket deleted**

2. p.12 l.331 Consistency of compound description, is there a convention for use of (((..)))
   or ({[(…)]}) ? – **now p. 11 l. 277 - Supplier lists the compound in his catalogue like this.**
   **The brackets are correct.**
   **Consistency of compound description, please check here:**

   https://www.acdlabs.com/iupac/nomenclature/93/r93_61.htm

3. p. 32 l.948 Zhenz or Zheng? – **now p. 31 l. 875 – name corrected to Zheng**

4. p.34 l.1038 J or JJ Zheng?  - **now p. 33 l. 958 – correct is Zheng J**

5. p.34 l.1060 Shuker, duplicate ref. – **now p. 33 – duplicate reference deleted**

6. p.35 l.1105 M Zhang, duplicate ref. scaffoldproteins or scaffold proteins? – **now p. 34 -**
   **duplicate reference deleted**